# Obesity-Related Kidney Disease: Current Understanding and Future Perspectives

**DOI:** 10.3390/biomedicines11092498

**Published:** 2023-09-09

**Authors:** Frederik F. Kreiner, Philip Andreas Schytz, Hiddo J. L. Heerspink, Bernt Johan von Scholten, Thomas Idorn

**Affiliations:** 1Novo Nordisk A/S, DK-2860 Søborg, Denmark; frfk@novonordisk.com (F.F.K.); ppsy@novonordisk.com (P.A.S.); bjos@novonordisk.com (B.J.v.S.); 2Department of Clinical Pharmacy and Pharmacology, University of Groningen, 9700 RB Groningen, The Netherlands; h.j.lambers.heerspink@umcg.nl

**Keywords:** obesity, overweight, chronic kidney disease, obesity-related kidney disease, fatty kidney disease

## Abstract

Obesity is a serious chronic disease and an independent risk factor for the new onset and progression of chronic kidney disease (CKD). CKD prevalence is expected to increase, at least partly due to the continuous rise in the prevalence of obesity. The concept of obesity-related kidney disease (OKD) has been introduced to describe the still incompletely understood interplay between obesity, CKD, and other cardiometabolic conditions, including risk factors for OKD and cardiovascular disease, such as diabetes and hypertension. Current therapeutics target obesity and CKD individually. Non-pharmacological interventions play a major part, but the efficacy and clinical applicability of lifestyle changes and metabolic surgery remain debatable, because the strategies do not benefit everyone, and it remains questionable whether lifestyle changes can be sustained in the long term. Pharmacological interventions, such as sodium-glucose co-transporter 2 inhibitors and the non-steroidal mineralocorticoid receptor antagonist finerenone, provide kidney protection but have limited or no impact on body weight. Medicines based on glucagon-like peptide-1 (GLP-1) induce clinically relevant weight loss and may also offer kidney benefits. An urgent medical need remains for investigations to better understand the intertwined pathophysiologies in OKD, paving the way for the best possible therapeutic strategies in this increasingly prevalent disease complex.

## 1. Introduction

Obesity and chronic kidney disease (CKD) are closely interlinked. Obesity is viewed as an independent risk factor for CKD, irrespective of diabetes, hypertension, and other well-established risk factors [1]. Accordingly, the concept of obesity-related kidney disease (OKD) has been introduced, reflecting that excess body weight may lead to CKD or the worsening of existing CKD, which is associated with premature mortality [2,3]. Recent global data suggest that in 2020, 988 million people above the age of five were living with obesity [4]. By 2030, it is predicted that 1.5 billion people, or 20% of the global population above the age of five, will have a BMI of 30 kg/m^2^ or more [4]. Almost half (44%) of the United States population with CKD also have obesity [5], and it has been projected that CKD in 2040 will become the fifth most common cause of death, in part because of the high and increasing prevalence of obesity [6].

A detailed understanding of OKD as a distinct disease remains to be fully established. This includes elucidating the interplay between OKD and cardiometabolic disease, the contribution of obesity-related mediators relative to other pleiotropic factors such as genetic susceptibility, overall disease burden in general, and the presence of comorbidities like diabetes and hypertension. Establishing such insights will help clarify the potential need for dedicated non-pharmacological and pharmacological treatments for OKD.

In this article, we review the current understanding of OKD and focus on the intertwined cardiometabolic abnormalities, which, in addition to obesity and diabetes, include risk factors such as hypertension, dyslipidemia, and systemic inflammation. We aim to provide an updated perspective on the bidirectional connections between impaired kidney function and obesity while discussing current pharmacotherapeutic options with a focus on novel opportunities for weight management. Finally, we provide thoughts on current and future developments that may help bridge existing knowledge gaps within OKD.

## 2. Obesity and CKD Definitions

Obesity, characterized by the accumulation of excess adipose tissue [7] and defined according to the WHO as a standalone disease [8], is usually categorized according to body mass index (BMI). A BMI of 25–30 kg/m^2^ is defined as overweight, while a BMI of at least 30 kg/m^2^ is the threshold for obesity in most circumstances [7]. In children and adolescents, BMI ranges according to age- and sex-specific percentiles are used [9]. Although BMI remains widely used, the distribution of fat across body compartments such as the subcutis and intra-abdominally may, in many cases, be a better measure [10,11]. Waist circumference is often used to gauge visceral adiposity, which is especially associated with a poor prognosis [10,12].

CKD is a serious and increasingly prevalent condition associated with severe morbidity and premature death, primarily in adults, worldwide [1]. The global prevalence of CKD is estimated to be approximately 9–10% [13] or higher [14], with many cases remaining undiagnosed. Combined with multiple direct and indirect implications in terms of management and costs [13,14], the high prevalence of CKD makes the disease a significant healthcare challenge, expected to be the fifth most common cause of death by 2040 [3]. CKD is a chronic progressive condition, often graded into CKD stages 1–5, according to kidney function (glomerular filtration rate [GFR], which is often estimated as eGFR) [1] and kidney damage stage A1-3, according to the degree of albuminuria (normal to mildly increased, moderately increased, or severely increased). Albuminuria is often an early sign of kidney damage and is associated with increased risk for CKD progression [15,16]. Among the five stages of CKD, stage 5 is kidney failure with an eGFR below 15 mL/min/1.73 m^2^, most often necessitating chronic kidney replacement therapy (chronic maintenance dialysis or kidney transplantation). The GFR declines by approximately 1 mL/min per year due to physiological aging [17]; other chronic diseases such as obesity may accelerate this rate, causing the faster depletion of residual kidney function [18,19,20]. A recent study showed that in people with CKD, rapid decline in eGFR is associated with deteriorating health-related quality of life [21].

## 3. Obesity-Related Kidney Disease

Several studies have found that obesity is associated with CKD. In a meta-analysis encompassing a broad range of studies, obesity (but not overweight) was associated with low eGFR (<60 mL/min/1.73 m^2^) and increased albuminuria in adults with BMI > 30 kg/m^2^ (>25 kg/m^2^ in Asians) compared with those with a BMI < 30 (<25 kg/m^2^ in Asians) (relative risk 1.36; 95% CI: 1.18–1.56) [22]. The same meta-analysis found that each incremental 1-kg/m^2^ increase in BMI was associated with a 2% greater incidence of progression to eGFR < 60 mL/min/1.73 m^2^ (relative risk of 1.02; 95% CI: 1.01–1.03) [22]. Based on 2011–2014 NHANES data, almost half (44.1%) of those with CKD also had obesity (BMI > 30 kg/m^2^) [5], compared to an overall obesity prevalence in the same dataset of around 38%. In the global population, around 10% have CKD [3], but a robust estimate of how many of these cases that can be attributed to obesity is not available, to our knowledge. However, individuals with obesity face a statistically significantly higher lifetime CKD risk (across CKD stages 1–5) compared with people with normal weight (41.0% vs. 32.5%, respectively) according to US estimates based on NHANES data [23]. Among several other comorbidities, obesity has been associated with a marked increased risk of kidney cancer [24].

Reflecting the multidirectional relationships among cardiometabolic risk factors, it remains to be clarified whether obesity is a trigger that causes CKD or a contributing factor, or both, in the development of CKD. Ultimately, and regardless of the underlying cause of CKD, the excretory and homeostatic functions of the kidneys may decline to an extent where kidney replacement therapy (dialysis or kidney transplantation) becomes necessary [1]. However, obesity may limit the chance of being listed as a candidate for kidney transplantation given the surgical challenges, profound scarcity of suitable donors, and negative impact of obesity on the chance of a successful outcome, although a recent report suggested a clear long-term benefit in people with as well as without obesity, between whom the transplantation success and survival did not appear to differ [25,26]. Nevertheless, there is a dire need for efficacious interventions that, when applied in a timely manner, help to prevent or slow the obesity-related accelerated progression of declining kidney function before any invasive replacement strategies become necessary.

Whereas the contribution of obesity in the development of CKD remains to be fully clarified, the pathophysiology of CKD in general has become relatively well described.

### 3.1. Pathophysiology of Chronic Kidney Disease

Loss of or damage to the functional units of the kidney, the nephrons, is the central pathological hallmark of CKD [1]. In the nephron, the podocyte is a principal cell type of the glomerular filtration barrier, which, in a healthy kidney, selectively filters the blood for a range of macromolecules [27,28]. When the barrier is compromised due to lost or damaged podocytes, macromolecules such as albumin eventually leak into the urine, causing albuminuria [27,28]. Podocytes are highly specialized epithelial cells that have become terminally differentiated and, therefore, no longer divide; accordingly, lost or malfunctioning podocytes cannot be replaced. Thus, to maintain the integrity of the glomerular filtration barrier, podocytes undergo hypertrophy, thereby initiating a vicious circle leading to further podocyte decline [29]. Of note, when increased amounts of albumin leak through the glomerular basement membrane, the resulting exposure of epithelial cells in the tubulointerstitium to high concentrations of albumin promotes a pro-inflammatory response, which can lead to interstitial inflammation and damage and, ultimately, nephron dysfunction [30].

To describe the damage to the renal glomeruli associated with excess body weight, the concept of obesity-related glomerulopathy (ORG) has been introduced [31]. ORG may be a major pathogenic mediator in OKD and can be seen as a histopathological manifestation of OKD [32,33]. A central component of ORG is podocyte damage and depletion [34]; secondary to this, glomerulosclerosis may develop as part of a complex of intra-kidney abnormalities that also include glomerulomegaly and mesangial expansion, secondary to increased blood flow and hyperfiltration [35].

Selected key aspects of the pathophysiology of OKD and CKD are illustrated in Figure 1, alongside established or potential pharmaceutical or non-pharmaceutical interventions.

### 3.2. Visceral Adiposity and the Fatty Kidney

In addition to the kidney-specific determinants of the susceptibility toward and severity of kidney disease in relation to obesity, various factors specific to excess adipose tissue also play a role. The distribution of excess adipose tissue across the body is one example [36]. As outlined above, whereas evidence suggests that subcutaneous fat tissue has relatively little pathogenic potential in the context of low amounts of visceral fat, visceral adiposity has been especially implicated in poor outcomes [36,37].

Visceral adiposity refers to the accumulation of fat within the abdominal cavity, specifically surrounding the internal organs [36]. This accumulation of adipose tissue is a major risk factor for cardiometabolic disease and can lead to kidney disease, directly or by exacerbating other risk factors with which visceral fat is closely interlinked. Excess visceral fat may accumulate as ectopic fat inside and outside the kidneys, similar to the buildup of fat in the liver in non-alcoholic fatty liver disease (NAFLD) and the advanced version thereof, NASH (non-alcoholic steatohepatitis) [38]. Kidneys characterized by widespread accumulations of adipose tissue are often referred to as fatty kidney disease (FKD) [39,40,41], a concept that is increasingly being recognized as central to poor kidney function in general, and in particular in relation to obesity [42] and T2D [42,43]. The seminal Framingham Heart Study found that intrarenal fat is strongly associated with the presence of visceral fat and systemic hypertension, as well as with albuminuria and impaired kidney function [37].

Regardless of the relative importance of other culprits such as diabetes, excess intra- and extrarenal fat tissue has specific negative consequences that can directly or indirectly cause or exacerbate poor kidney function. As suggested by García-Carro and colleagues, it is valuable to consider three main interrelated pathways as those that drive the negative consequences of obesity on the kidneys [44]: the adiposity pathway, the hemodynamics pathway, and a pathway involving the consequences of insulin resistance and hyperinsulinemia.

### 3.3. Proinflammatory Hormones and Lipotoxicity: The Adiposity Pathway

#### 3.3.1. Adipokines

Adipose tissue, and visceral fat in particular, is endocrinologically active, producing hormones (i.e., adipokines) such as adiponectin, leptin, resistin, and visfatin, as well as proinflammatory cytokines, e.g., tumor necrosis factor (TNF)-α and interleukin (IL)-6 [36,45]. Although the mechanisms of action and relative contributions of these hormones are complex and incompletely understood, the adipokines and cytokines are known to be intimately implicated in the development of inflammation, aberrant lipidemia, oxidative stress, and insulin resistance, as well as disturbed hemodynamics due to activation of the renin-angiotensin-aldosterone system (RAAS) [36,45].

In obesity, blood levels of adiponectin are decreased, whereas leptin levels are increased [46,47]. It is well established that adiponectin levels are inversely related to insulin resistance and glucose and fatty acid metabolism [36,46]. Accordingly, the adiponectin receptor is found on all cell types within the glomeruli, including the podocytes, illustrating that this adipokine potentially has several actions in the kidney [48]. In mice, a lack of adiponectin has been shown to negatively affect podocyte function [49]. Furthermore, adiponectin can act as antifibrotic in several organs [36,45], with accumulating evidence supporting such effects in the kidneys as well [50].

Leptin, a key regulator of food intake and energy homeostasis [51], increases proportionally with adipose tissue [47]. In CKD, systemic leptin levels have been shown to be elevated [52]; however, it remains unclear whether this observation is secondary to reduced renal function, as leptin is cleared renally. Among other actions, leptin has been shown to promote hypertension through the activation of the sympathetic nervous system [53], and the adipokine also increases fatty acid oxidation, leading to further increases in oxidative stress, as well as to the secretion of proinflammatory cytokines [47,54,55].

#### 3.3.2. Inflammation and Lipotoxicity

Secretion by adipose tissue in and around the kidneys of TNF-α, IL-6, and other cytokines and chemokines results in the recruitment of macrophages [56] and other effector cells. In combination with other processes and depending on the stage of CKD, this recruitment of immune cells can initiate or exacerbate a proinflammatory milieu [36,45,56] in and around the kidneys that may already be present to some extent due to aging-related immune system dysfunction, contributing, alongside obesity, to chronic low-grade inflammation [57,58]. Consequently, the production of reactive oxygen species (ROS) increases, leading to oxidative stress and, in turn, damage to the renal tubuli, resulting in abnormal ion transport and other functional disturbances [59,60]. Furthermore, renal fibrosis may develop due to the inflammatory process in general but also due to the inflammation-mediated enhanced ROS-induced promotion of profibrotic compounds such as transforming growth factor-β (TGF-β1) [61] and plasminogen activator inhibitor-1 (PAI-1) [62,63]. Of note, TGF-β1 also possesses anti-inflammatory properties [61], illustrating the complexity of the processes involved in the pathogenesis of CKD. Nevertheless, renal fibrosis, regardless of the cause, may initiate a vicious circle where the rate of nephron loss is accelerated. In addition, the ROS burden may also stress the endoplasmic reticulum (ER), leading to abnormal protein folding [64].

Fatty acids have been shown to promote systemic oxidative and ER stress [65], which is a result of the toxicity of lipid accumulation in obesity. The renal consequences of this lipotoxicity include functional and structural changes to, for example, the podocytes and tubules [66].

Moreover, peroxisome-proliferator-activated receptor gamma (PPARγ), which is intimately involved in a range of metabolic pathways, including glucose metabolism, energy homeostasis, adipogenesis, and triglyceride handling [67,68], is believed to be central to the function of the kidneys [69], and it is expressed throughout the kidney. For example, in line with how PPARγ has been implicated in the pathophysiology of obesity and type 2 diabetes [67,68,70,71], mutations in the gene encoding PPARγ may impact the severity or susceptibility of OKD and CKD in general [69,72]. Overall, a kidney-protective effect of PPARγ has long been proposed [69], with one report suggesting that this beneficial effect is driven at least partly by reduced renal fibrosis via inhibition of TGF-β1 [73].

### 3.4. RAAS Activation and Intraabdominal Pressure: The Haemodynamics Pathway

The endocrine activity of excess adipose tissue, whether located in the abdomen or intra-renally, may lead to increased intra-renal blood pressure [74], often referred to as intraglomerular hypertension, as a result of the activation of the RAAS, as outlined above [75,76]. The activation of the RAAS may also lead to the increased reabsorption of sodium in the tubules [77], contributing to systemic as well as intraglomerular hypertension. In addition, because of increased mass in the abdomen, visceral adiposity causes increased intra-abdominal pressure, leading to renal compression [76,78]. This may impact renal blood flow and perfusion, and lead to further intraglomerular hypertension [76]. Intraglomerular hypertension causes shear-related stress on the renal vasculature and the glomerular cells, such as podocytes [76,79], eventually contributing to declining kidney function. Lastly, obesity is associated with the increased activation of the sympathetic nervous system [76,80] and with modulation of the gastro–renal axis [76], both of which together may also lead to systemic and/or intraglomerular hypertension.

### 3.5. Insulin Resistance and Hyperinsulinemia

Insulin resistance in general and in the kidney specifically also plays an important role in the development of kidney damage. Insulin resistance in the setting of obesity is predominantly caused by the overexpression of proinflammatory cytokines released by adipose tissue, as discussed above. These cytokines may negatively impact insulin signaling [81,82]. Normal kidney function is critically dependent on insulin signaling [83], and insulin directly acts on several parts of the nephron, thereby playing a role in modulating kidney functions such as the glomerular filtration barrier via the modulation of podocytes [84]. In animal models, as well as in in vitro studies with cultured podocytes, insulin has been shown to increase the permeability of the glomerular barrier to albumin [85], and insulin-resistant podocytes due to, for example, hyperglycemia or hyperinsulinemia are associated with albuminuria [86]. Furthermore, insulin has been shown to promote tubulointerstitial fibrosis [87].

## 4. Current Therapeutic Options in Obesity and Chronic Kidney Disease

No dedicated treatment options or treatment guidelines are yet available for OKD, and current treatment strategies based on both non-pharmacological interventions and pharmacotherapies target obesity and CKD individually. However, some treatment options are beneficial across the two conditions. Established or potential pharmacotherapies and non-pharmacological interventions that may address the pathophysiological characteristics of OKD are summarized in Figure 1.

### 4.1. Non-Pharmacological Treatment

Non-pharmacological treatment strategies play a central role in managing CKD and obesity, individually or as comorbidities, and typically involve lifestyle changes such as dietary modifications and increased physical activity [42,88,89,90]. These changes aim to alleviate symptoms and prevent the diseases from further progression.

In managing CKD and obesity, a combination of lifestyle interventions, medications, and bariatric surgery may be considered. Although clinical trials have suggested the potential renal and cardiovascular benefits of lifestyle modifications in people with pre-existing CKD, the superiority of different dietary regimens to facilitate weight loss in CKD is still unclear. Additionally, while bariatric procedures are associated with a lower risk of end-stage kidney disease [91], they may also increase the risk of acute kidney injury and other complications, especially in people with diabetes and/or obesity [92]. Further research is needed to refine these strategies and improve the non-pharmacological management of CKD and obesity.

### 4.2. Pharmacotherapies

With the understanding that obesity and CKD are highly interrelated and often coincide, medicines that address one or both conditions are highly desirable. RAAS blockers (ACE (angiotensin-converting enzyme) inhibitors or ARBs (angiotensin II receptor blockers)) are recommended in all patients with CKD and albuminuria [16]. As discussed below, sodium–glucose cotransporter-2 (SGLT-2) inhibitors, and more recently non-steroidal mineralocorticoid receptor antagonists (nsMRAs), have been introduced as pharmacotherapeutic agents in CKD with (SGLT-2 inhibitors and nsMRAs) or without concurrent T2D (SGLT-2 inhibitors). However, while SGLT-2 inhibitors can reduce the risk of kidney failure and cardiovascular events and provide improvements in glycemic control, neither this drug class nor (ns)MRAs markedly address obesity per se. As summarized in a later section, medicines based on glucagon-like peptide-1 (GLP-1) are available for weight management in people with overweight or obesity and for the management of T2D, and evidence suggests potential kidney-protective benefits as well. There is hope that these novel drug classes can help address the persisting and pronounced unmet medical need for pharmacotherapeutic options that specifically target OKD.

#### 4.2.1. SGLT-2 Inhibitors

SGLT-2 inhibitors, such as empagliflozin, canagliflozin, and dapagliflozin, function by preventing the reabsorption of sodium and glucose in the kidneys, thereby promoting their excretion in urine. While SGLT-2 inhibitors provide the lowering of blood glucose levels [93], their effect on natriuresis is transient and modest [94]. These pleiotropic agents were initially used for managing T2D, but emerging evidence has confirmed their beneficial effects on cardiovascular and kidney outcomes, even though the mechanistic basis for these benefits remains incompletely understood [94,95]. Studies have shown that SGLT-2 inhibitors reduce the risk of major adverse cardiovascular events, hospitalization for heart failure, and the progression of CKD [96,97,98,99,100,101,102,103,104,105,106,107,108,109,110,111]. In CKD, they slow the decline in glomerular filtration rate, reduce albuminuria, and delay the onset of end-stage kidney disease [96,97,98,99,100,101,102,103,104,105,106,107,108,109,110,111]. SGLT-2 inhibitors can promote weight loss in individuals with T2D via the increase in the amount of glucose excreted in the urine, which results in a loss of calories [112]. However, the weight loss associated with SGLT-2 inhibitors is usually modest [113]. Especially in trials including CKD patients, weight loss with SGLT-2 inhibitors has been minimal [114]. Accordingly, while SGLT-2 inhibitors offer kidney protection in people with or without obesity, this benefit appears unrelated to weight reduction. Dual SGLT-1 and SGLT-2 inhibitors might produce more pronounced weight loss [115], and were recently approved to reduce the risk of cardiovascular death or worsening of heart failure in people with pre-existing heart failure or with T2D, CKD, and cardiovascular risk factors [95].

#### 4.2.2. Non-Steroidal Mineralocorticoid Receptor Antagonists

nsMRAs represent an emerging therapeutic option that improves cardiorenal outcomes in people with CKD and diabetes, especially in combination with other medical therapies used in heart failure and CKD. nsMRAs like finerenone exhibit a risk for hyperkalemia [116], although lower than that with steroidal MRAs [117]. In people with CKD and T2D, nsMRA use resulted in a reduction in the risks of major adverse cardiovascular events [118] and diabetic kidney disease progression [119], albeit with no impact on bodyweight.

#### 4.2.3. GLP-1-Based Medicines

GLP-1 is one of the two major incretin hormones. GLP-1 is secreted from the intestines upon food intake and plays a critical role in regulating blood glucose levels by enhancing insulin release and reducing glucagon secretion [120]. It also promotes satiety by acting on brain receptors, a property that has been harnessed pharmacologically to reduce body weight [120]. GLP-1 receptor agonists (GLP-1 RAs) are dosed once-daily or once-weekly and are indicated for weight management in those who are overweight (with ≥1 weight-related comorbidity) or obese (specifically, semaglutide and liraglutide) and for improving glycemic control in people with T2D (all marketed GLP-1 RAs) [120,121,122]. Semaglutide and dulaglutide also reduce cardiovascular risk in people with T2D [123,124,125,126].

Newer-generation GLP-1 RAs, such as high-dose once-weekly injectable semaglutide (up to 2.0 mg in T2D and 2.4 mg in weight management) [127,128], once-daily oral semaglutide (up to 50 mg in weight management [129]) and the GLP-1/GIP dual RA tirzepatide [130,131,132], have demonstrated significant reductions in glycated hemoglobin and body weight. A fixed-dose combination drug, the combination of semaglutide and the amylin analogue cagrilintide (CagriSema), is currently under clinical testing for use in T2D [133] and weight management [134]. Furthermore, retatrutide, a GLP-1/GIP/glucagon triagonist, is also in development for glycaemic control and weight management, and has shown marked weight-lowering effects in relatively short-term studies [135]. The longer-term efficacy and safety of these new incretin-based therapeutics are being investigated.

While no GLP-1 based medicines are currently indicated to improve kidney outcomes, recent meta-analyses suggest a kidney-protective potential [125,136]. In addition, a post-hoc analysis exploring the effect of semaglutide on albuminuria and kidney function based on data from the STEP 1, 2, and 3 trials with once-weekly semaglutide 2.4 mg for weight management in people who are overweight/obese (and T2D in STEP 2) was recently reported [137]. Across the trials, semaglutide did not impact change in eGFR vs. placebo, reflecting that most trial participants had normal kidney function at enrolment. Conversely, in STEP 2 (the only one of the three trials where albuminuria was measured) in people with T2D and overweight or obesity, semaglutide markedly improved albuminuria, measured as the urinary albumin-to-creatinine ratio (UACR), which at week 68 was reduced statistically significantly by 32.9% with semaglutide 2.4 mg vs. placebo (*p* = 0.003). In STEP 2, the effect of semaglutide on UACR appeared to be dose dependent [138] and was more pronounced in participants with macro- or microalbuminuria compared with those normoalbuminuria, but otherwise was consistent across subgroups, including by baseline BMI, eGFR, glycated hemoglobin, systolic blood pressure, and, clinically relevantly, SGLT-2 inhibitor or RAAS inhibitor use. The analyses corroborate previous findings of an albuminuria-reducing effect of semaglutide [123].

A dedicated kidney outcomes trial with semaglutide (FLOW, ClinicalTrials.gov ID: NCT03819153) is ongoing, and exploratory trials are underway to understand the kidney-specific mechanism of action for semaglutide (REMODEL, ClinicalTrials.gov ID: NCT04865770; SMART, ClinicaTrials.gov ID NCT04889183) and tirzepatide (TREASURE-CKD, ClinicalTrials.gov ID: NCT05536804).

Given their efficacy in addressing hyperglycemia, hyperinsulinemia, and excess adipose tissue, GLP-1 based therapies are potentially valuable in preventing and mitigating obesity-related kidney disease. Furthermore, they have been shown to significantly reduce blood pressure [123,139,140,141,142,143] and inflammation [144,145,146,147], two other key factors in the pathophysiology of obesity-related kidney disease.

## 5. Future Perspectives and Challenges

As outlined above, there is an urgent need for innovative and targeted approaches toward further understanding, diagnosing, and managing OKD. Nonetheless, achieving this ambition is impeded by several knowledge gaps that must be bridged to augment our comprehension and therapeutic strategies. First, from a diagnostic perspective, further development of methods for measurement and estimation of kidney function in people with obesity and during weight loss is warranted. Such methods may go beyond creatinine-based or cystatin C-based eGFR equations, and how to best adjust for body surface area (BSA), including standardized vs. individualized BSA, within obesity should be clarified. Second, the development of biomarkers, including diagnostic, prognostic, and predictive biomarkers, holds potential for improved and targeted therapies aiming for a precision-medicine approach. This enables the diagnosis of people with OKD followed by risk stratification for disease progression and, ultimately, identification of responders to treatment. Urinary and blood-based circulating biomarkers beyond eGFR and UACR, in addition to novel imaging or genetic markers, will be important to focus on for future improvements in diagnosing, grouping, and treating OKD/FKD/ORG. Finally, pivotal, as well as mechanistic, trials specifically targeting people with CKD and obesity are warranted, and several emerging therapies have already shown promising, preliminary results on clinically relevant CKD outcomes within this patient segment. Similarly, preclinical and clinical studies are warranted to better understand the potential kidney-protective effect of weight loss per se, and weight loss achieved by pharmacotherapies, where direct kidney-specific effects, as well as indirect effects (e.g., weight loss, blood pressure reduction, and more), may play a role in the treatment of OKD. This may further assist in guideline development.

In conclusion, these perspectives underscore the significance of a multi-faceted approach in addressing obesity-related CKD (i.e., OKD). From diagnostic methods to therapeutic interventions, every facet requires attention to augment our understanding and management of this pressing health issue.

eGFR, estimated glomerular filtration rate; GLP-1, glucagon-like peptide-1; RAAS, renin-angiotensin-aldosterone system; SGLT-2, sodium-glucose cotransporter-2; inhibitors

## Figures and Tables

**Figure 1 biomedicines-11-02498-f001:**
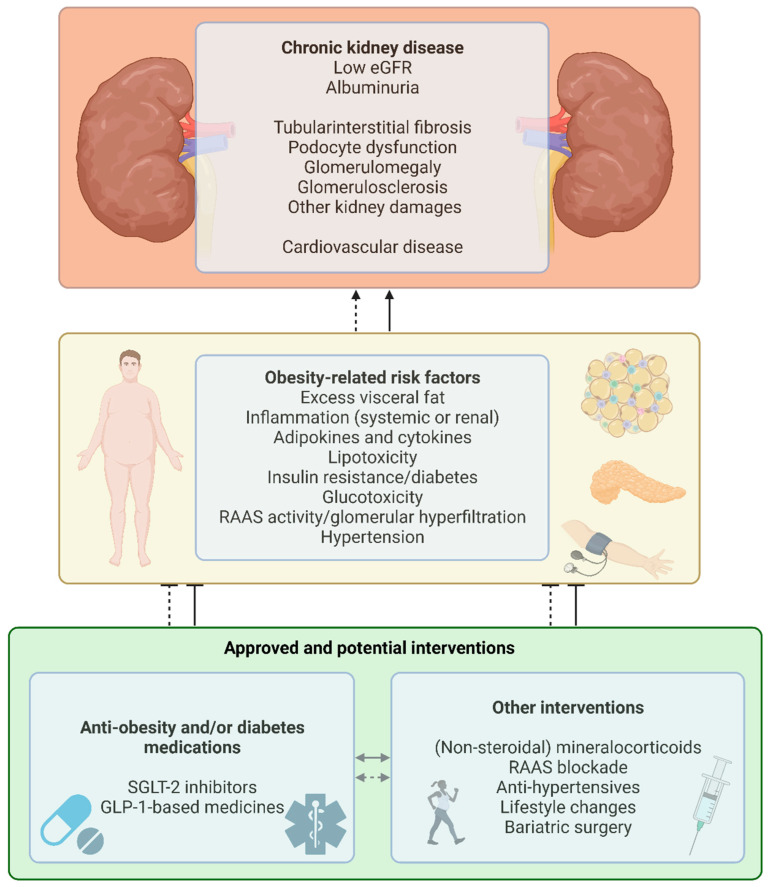
Pathophysiological associations and therapeutic options in obesity-related kidney disease. Figure 1 shows the known or suspected obesity-related risk factors (yellow box) directly or indirectly implicated in the development of chronic kidney disease (CKD). CKD is traditionally categorized according to the presence and degree of albuminuria and glomerular filtration capacity (eGFR), and by a range of different kinds of kidney-specific damage, as well as resulting or exacerbating cardiovascular disease (red box). The green box shows approved or suggested/potential pharmaceutical or non-pharmaceutical interventions that may address obesity-related risk factors in CKD. Dashed and solid lines indicate potential and well-established associations/effects, respectively. Details are available in the text. Made with BioRender.com.

## Data Availability

Not applicable.

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
