# Peer review of "Obesity-Related Kidney Disease: Current Understanding and Future Perspectives"

_biomedicines, 2023, doi:10.3390/biomedicines11092498_

Round 1

Reviewer 1 Report

This is a timely and thorough review of obesity-related kidney disease (OKD).  Currently, there are no designated treatments for OKD.  The authors present a case for the urgent need for a better understanding of the intertwined pathophysiologies in OKD for non-pharmacologic and pharmacologic therapeutic strategies.  The authors describe the OKD and the current therapies for treating CKD or obesity, highlighting the few studies that have examined the effects of treatments on obese patients. 

Author Response

We thank Reviewer 1 for the positive reception of our manuscript - much appreciated.

Reviewer 2 Report

The manuscript is interesting, summarizing obesity as a risk factor for CKD.

These are my concerns

There are typo errors and grammar that should be corrected.

The main idea is okay, and the order is easy to follow; however, some parts are very superficial, or the ideas are incomplete for example:

As suggested by García-Carro and colleagues, it is valuable to consider three main interrelated pathways that drive the negative consequences of obesity on the kidneys ¿What are the adverse effects? Which mechanisms are involved?

 These cytokines may negatively impact insulin signaling  [80,81]. Normal kidney function is critically dependent on insulin signaling [82], and insulin directly act on several parts of the nephron, thereby playing a role in modulating kidney functions such as glomerular filtration barrier via modulation of podocytes [83]. ¿What are the kidney functions? Some sections are vast and well described as GLP1, and others very poor as SLGT2 or Non-steroidal mineralocorticoid receptor antagonists.

In general, the manuscript needs a deeper description of the mechanisms.

None

Author Response

The manuscript is interesting, summarizing obesity as a risk factor for CKD.

Thank you.

These are my concerns

There are typo errors and grammar that should be corrected.

Thank you. We have carefully checked for typos and grammar mistakes and revised accordingly.

The main idea is okay, and the order is easy to follow; however, some parts are very superficial, or the ideas are incomplete for example:

As suggested by García-Carro and colleagues, it is valuable to consider three main interrelated pathways that drive the negative consequences of obesity on the kidneys ¿What are the adverse effects? Which mechanisms are involved?

We thank the reviewer for this constructive comment and agree with the reviewer that the pathways and mechanisms in question are indeed important topics to cover in this review. However, we respectfully note that the details related to the three pathways in question are indeed described in the sections 3.3.1, 3.3.2, 3.4, and 3.5. In our view, adding additional details is beyond the limits of the intended scope and length of the manuscript. Yet, if the reviewer maintains that additional details in this manuscript are needed, we will of course be happy to reassess.

These cytokines may negatively impact insulin signaling  [80,81]. Normal kidney function is critically dependent on insulin signaling [82], and insulin directly act on several parts of the nephron, thereby playing a role in modulating kidney functions such as glomerular filtration barrier via modulation of podocytes [83]. ¿What are the kidney functions? Some sections are vast and well described as GLP1, and others very poor as SLGT2 or Non-steroidal mineralocorticoid receptor antagonists.

In general, the manuscript needs a deeper description of the mechanisms.

We thank the reviewer for this constructive comment. As indicated in the cited text, by “kidney function”, we refer to the filtration barrier. We do not, however, intent the manuscript to elaborate comprehensively on all aspects of CKD (OKD) but aim to provide an overview of the most significant mechanisms (according to the current understanding) connecting CKD and the prominent metabolic derangements characterising obesity such as insulin resistance. Accordingly, we deliberately decided not to go into too many speculative details with regards to, for example, how insulin signalling and hyperinsulinemia may compromise kidney function, because we believe that this aspect is less well understood.

Regarding the description of SGLT-2 inhibitors and their therapeutic potential in OKD, we have now updated this section with additional details as suggested, including a mention of dual SGLT-1 and SGLT-2 inhibitors. Due to space constraints, we decided not to elaborate on the section of nsMRAs, also considering that we view this an emerging option, even though we fully recognize its potential in cardiorenal disease. However, to our knowledge, there appear to be little or no weight-loss benefit associated with, for example, finerenone, potentially limiting its relevance in CKD associated with excess body weight.

Reviewer 3 Report

1. Lines 37-38: please change the wording to suit better how obesity is diagnosed during childhood and adolescence (a BMI over 30 kg/m2 is not the referral used in paediatric patients, and the source reference acknowledges this fact).

2. Due to a typing error, there are two 3.2 subsections in the text (lines 135 and 162). Please correct.

3. The text includes a few examples of a second typing variant (“SGLT2 inhibitors”) besides the most used one (“SGLT-2 inhibitors”). Please uniformise throughout the manuscript.

4. Lines 282-283: SGLT-2 inhibitors also prevent sodium, not just glucose reabsorption, in the kidneys. This mechanism supposedly has a significant role in the cardiorenal protective effects. Other pleiotropic actions of SGLT-2 inhibitors should also be briefly acknowledged.

5. Dual SGLT-1 and SGLT-2 inhibitors are nowadays tested for a potentially more pronounced weight loss effect than current class representatives. The authors should perhaps briefly comment on the potential cardiorenal benefits of such a drug, even though no clinical studies currently test this hypothesis.

6. Is the current position of Figure 1 in the manuscript best suited to reflect the importance of its content appropriately? The authors should at least use a sixth section of Conclusions, with the significance of Figure 1 acknowledged within the text.

7. The quality of the English language is generally fine. In the Abstract section, the phrase in lines 19-21 may perhaps become more logical if changed to “Pharmacological interventions such as sodium-glucose co-transporter 2 inhibitors and a non-steroidal mineralocorticoid receptor antagonist finerenone provide kidney protection but have limited or no impact on body weight”.

Author Response

Comments and Suggestions for Authors

  1. Lines 37-38: please change the wording to suit better how obesity is diagnosed during childhood and adolescence (a BMI over 30 kg/m2 is not the referral used in paediatric patients, and the source reference acknowledges this fact).

Ad point 1: Thank you. We have included a sentence clarifying this important fact in the first paragraph under section 2.0: “Obesity and CKD definitions”:

“In children and adolescents, BMI ranges according to age- and sex-specific percentiles are used [9].”

  1. Due to a typing error, there are two 3.2 subsections in the text (lines 135 and 162). Please correct.

    Ad point 2: Thank you for pointing out this error. We have updated accordingly.
  2. The text includes a few examples of a second typing variant (“SGLT2 inhibitors”) besides the most used one (“SGLT-2 inhibitors”). Please uniformise throughout the manuscript.

Ad point 3: Thank you. We have updated accordingly and aligned to “SGLT-2”, except in the reference list where we have kept the style of the cited papers.

  1. Lines 282-283: SGLT-2 inhibitors also prevent sodium, not just glucose reabsorption, in the kidneys. This mechanism supposedly has a significant role in the cardiorenal protective effects. Other pleiotropic actions of SGLT-2 inhibitors should also be briefly acknowledged.
  2. Dual SGLT-1 and SGLT-2 inhibitors are nowadays tested for a potentially more pronounced weight loss effect than current class representatives. The authors should perhaps briefly comment on the potential cardiorenal benefits of such a drug, even though no clinical studies currently test this hypothesis.

Ad points 4 and 5: Thank you for these relevant suggestions. We have revised the section on SGLT-2 inhibitors (end of section 4.2.1) to also include mentions of natriuresis and the SGLT-1 and SGLT-2 dual inhibitors, including the recent approval of sotagliflozin.

"Dual SGLT-1 and SGLT-2 inhibitors might produce more pronounced weight loss [115] and were recently approved to reduce the risk of cardiovascular death or worsening of heart failure in people with pre-existing heart failure or with T2D, CKD and cardiovascular risk factors [95]."

  1. Is the current position of Figure 1 in the manuscript best suited to reflect the importance of its content appropriately? The authors should at least use a sixth section of Conclusions, with the significance of Figure 1 acknowledged within the text.

Ad point 6: We appreciate this comment by Reviewer 3 and have now included a reference to the figure in section 3.1 (last sentence: Selected key aspects of the pathophysiology of OKD and CKD are illustrated in Figure 1, alongside established or potential pharmaceutical or non-pharmaceutical interventions.”) and section 4.0 (last sentence: Established or potential pharmacotherapies and non-pharmacological interventions that may address the pathophysiological characteristics of OKD are summarised in Figure 1.”)

Comments on the Quality of English Language

  1. The quality of the English language is generally fine. In the Abstract section, the phrase in lines 19-21 may perhaps become more logical if changed to “Pharmacological interventions such as sodium-glucose co-transporter 2 inhibitors and a non-steroidal mineralocorticoid receptor antagonist finerenone provide kidney protection but have limited or no impact on body weight”.

Ad point 7: Thank you; we have updated accordingly.